# Temporal Filtering to Improve Single Molecule Identification in High Background Samples

**DOI:** 10.3390/molecules23123338

**Published:** 2018-12-17

**Authors:** Alexander W. A. F. Reismann, Lea Atanasova, Lukas Schrangl, Susanne Zeilinger, Gerhard J. Schütz

**Affiliations:** 1Institute of Applied Physics, TU Wien, Getreidemarkt 9, A-1060 Vienna, Austria; reismann@iap.tuwien.ac.at (A.W.A.F.R.); lea.atanasova@boku.ac.at (L.A.); schrangl@iap.tuwien.ac.at (L.S.); 2Department of Microbiology, University of Innsbruck, Technikerstrasse 25, A-6020 Innsbruck, Austria; Susanne.Zeilinger@uibk.ac.at; 3Institute of Chemical, Environmental and Bioscience Engineering, TU Wien, Gumpendorferstrasse 1a, A-1060 Vienna, Austria

**Keywords:** single molecule microscopy, super-resolution microscopy, Fourier filter, background fluorescence, image processing

## Abstract

Single molecule localization microscopy is currently revolutionizing the life sciences as it offers, for the first time, insights into the organization of biological samples below the classical diffraction limit of light microscopy. While there have been numerous examples of new biological findings reported in the last decade, the technique could not reach its full potential due to a set of limitations immanent to the samples themselves. Particularly, high background signals impede the proper performance of most single-molecule identification and localization algorithms. One option is to exploit the characteristic blinking of single molecule signals, which differs substantially from the residual brightness fluctuations of the fluorescence background. To pronounce single molecule signals, we used a temporal high-pass filtering in Fourier space on a pixel-by-pixel basis. We evaluated the performance of temporal filtering by assessing statistical parameters such as true positive rate and false discovery rate. For this, ground truth signals were generated by simulations and overlaid onto experimentally derived movies of samples with high background signals. Compared to the nonfiltered case, we found an improvement of the sensitivity by up to a factor 3.5 while no significant change in the localization accuracy was observable.

## 1. Introduction

The last decade has seen a variety of microscopy methods which enable circumventing the classical diffraction limit of light microscopy [1]. One prominent set of techniques utilizes the stochastic switching of single molecules between a fluorescent “on-state” and a non-fluorescent “off-state”. In these single molecule localization microscopy (SMLM) approaches, the problem of resolving two or more point emitters was shifted to the problem of obtaining their correct position [2]. The single molecule localization precision is essentially determined by the brightness of the signal over background noise, but hardly depends on the width of the signal, therefore it allows for imaging structures at a resolution which is not limited by diffraction [3,4,5,6]. A related technique has been termed points accumulation for imaging in nanoscale topography (PAINT), and makes use of on/off signals upon binding of a fluorescent molecule to the biomolecule of interest [7,8].

Soon, however, researchers discovered practical problems when performing SMLM analysis. Solutions were presented, e.g., for heavily overlapping signals [9,10,11], and quantitative error assessments were also presented [12]. The general understanding is that images recorded under suboptimal conditions may yield numerous artifacts, partly because background fluorescence becomes problematic [13]. Not only does increased background fluorescence reduce localization precision [3], it may lead to the complete loss of signals, as well as the spurious detection of false positive signals. Background fluorescence particularly affects three-dimensional imaging of cells, as they cannot be recorded under total internal reflection fluorescence microscopy. In that case, out-of-focus fluorescence of dyes in the on-state in SMLM and unbound fluorophores in PAINT contribute to the overall background. 

Background fluorescence becomes especially problematic in samples showing high cellular autofluorescence [14,15,16]. Cellular autofluorescence is often of unknown origin, with various components contributing to the different spectral regions, including NADH, flavins, and lipofuscins. In many biological systems, autofluorescence decreases with increasing wavelength, making unambiguous single molecule detection more robust in the red region of the spectrum. In some cases, however, working with red dyes still gives insufficient results, or—in case of multi-color microscopy—is not an appropriate solution. Fungi represent one example of samples with high autofluorescence, which has even been proposed as means for categorizing different types of fungal pathogens [16]. In principle, it is possible to use the characteristic color of organic dyes for discrimination against autofluorescence [17], however, this approach assumes similar autofluorescence spectra across the sample, which a priori may not be the case. 

Here, we propose an approach that utilizes the fact that background fluorescence—both autofluorescence but also unbound fluorophores in PAINT—hardly fluctuates in time, whereas single molecules are driven into characteristic on–off cycles. Steps towards such scenarios have been published, e.g., by subtracting estimated background images that were constructed from time-averaged adjacent frames [18,19]. We extended this idea by performing high-pass filtering in Fourier space on a pixel-by-pixel basis, which allows for increasing the contrast between the single molecule signals and background. We quantified the improved performance of temporal filtering using simulated ground truth single molecule signals overlaid on recorded images from the filamentous fungus *Trichoderma atroviride*.

## 2. Results

Figure 1 shows the challenges which have to be tackled when applying SMLM to samples of high background fluorescence, such as filamentous fungi. Throughout this study we used *Trichoderma atroviride* (ATCC 74058) as a test sample. In Figure 1a we show a fluorescence image of an unstained wild-type cell. For comparison, in Figure 1b we show a cell expressing an *sfp2-mEGFP* fusion construct [20] stained with AlexaFluor 647 (AF647) conjugated GFP-Trap. The images were recorded under epi-illumination configuration using common dSTORM buffer (see methods). Apparently, the images were characterized by substantial background signal with similar or even higher magnitude as the single molecule brightness of AF647, which impedes the identification and localization of the single molecule signals from single frames. 

In the following, we propose a method that exploits the time domain in order to improve the contrast between single molecule signals and background fluorescence. We exploited the fact that background fluorescence hardly fluctuates between single frames and shows only marginal photobleaching during the whole imaging sequence. In contrast, single molecule signals blink from frame to frame. A pixel-by-pixel-based Fourier transform of the whole image sequence should be able to discriminate the two contributions—autofluorescence will mainly yield contributions to the low frequency part of the spectrum, whereas single molecule signals pop up in the high frequency part of the spectrum (Figure 2). The low frequency part of the spectrum up to a threshold ω_T_ is set to zero before the spectrum is back transferred to the time domain. In the following, we define ω_T_ in percent of the whole Fourier spectrum. To avoid negative values in the back-transformed image, we identified the lowest pixel value from the whole image sequence and added it to all pixels.

To assess the performance of the method, we simulated fluorescence signals as ground truth and overlaid them with movies of unlabeled *T. atroviride* wild-type hyphae measured under typical STORM conditions. Five separate movies were used as autofluorescence models (Figure 3). We varied the following parameters of the single molecule signals: the brightness, defined as the total number of counts per molecule B, and the mean duration of the on and off state (τ_on_ and τ_off_, respectively) of the single molecule blinking traces. We simulated single molecule emitters at densities ranging from 0.5 to 10 localizations per µm². The brightness was simulated following a lognormal distribution [21] with mean brightness values between 500 and 1000 counts per signal; this exceeds the standard deviation of the autofluorescence per pixel by a factor of 3 to 6. The width of the point spread function was chosen to match the data of AF647 (σ = 160 nm). The duration of single on and off periods was assumed to be exponentially distributed, with τ_on_ ranging between 1 and 21 frames; τ_off_ was varied between $20\cdot \tau_{on}$ and $100\cdot \tau_{on}$. In particular, this includes situations with very rare blinking events, with mean off-times of 2100 frames. Single molecule emitters were distributed randomly across the area covered by the fungus. The temporal Fourier filtered image sequences were finally analyzed with standard single molecule localization algorithms.

Figure 4 shows three exemplary images—out of a sequence of 10,000 images—of an unstained wild-type hypha, to which simulated single molecule signals were added. While in the raw data it is difficult to identify the single molecule signals (Figure 4a), they can be nicely discriminated against background upon temporal filtering (Figure 4b). A representative image of a single molecule signal before and after temporal filtering is shown in Figure 5. Note the improved signal to noise ratio on each image and the reduced signal fluctuations over time. 

Our method particularly strives to improve the true positive rate (TPR) of single molecule signals, which is defined as$\;\mathrm{TPR}\;=\;\frac{\mathrm{TP}}{\mathrm{FP}+\mathrm{TP}}\;$, with TP and FN denoting the number of true positive and false negative signals, respectively. In the ideal case of no missed signals (FN = 0), TPR would be equal one; on the other hand, if most signals were missed, TPR approaches zero. To quantitatively assess the new method, we took five representative movies from *T. atroviride* and added single molecules at various brightness values, blinking rates, and densities. Data were analyzed with and without temporal filtering using different thresholds $\omega_{T}$. In general, TPR increased substantially after temporal filtering, with up to 3.5-fold higher probability for correctly detecting a single molecule signal (Figure 6a). In Figure 6b we show the ω_T_ dependence of the mean TPR, which was obtained by averaging over all data obtained at the various simulation parameter settings. While for low ω_T_, the TPR approaches the value achieved without application of the temporal filter (here TPR ~0.5), it increases to an average of about 0.7. 

While TPR is sensitive to erroneously missed signals, it is also important to keep track of erroneously detected signals. Hence, the second parameter of interest here is the false discovery rate $\mathrm{FDR}\;=\;\frac{\mathrm{FP}}{\mathrm{FP}+\mathrm{TP}}$, which measures the amount of false positive signals related to all detected signals. In an ideal test FDR would be equal to zero, which is equivalent to the absence of false positive signals. The worst case scenario, on the other hand, is given by FDR = 1, which corresponds to only false positive signals. The red curve in Figure 6b shows the ω_T_-dependence of FDR: without temporal filtering (ω_T_ → 0) the false positives largely exceed the true positive signals, yielding FDR ~1; with increasing thresholds, FDR dramatically decreases to values below FDR = 0.5. 

An appropriate tool for assessing the performance of a binary classifier test is plotting the receiver operating characteristics (ROC). Usually, it shows TPR versus the false positive rate $\mathrm{FPR}\;=\;\frac{\mathrm{FP}}{\mathrm{FP}+\mathrm{TN}}$ for different test parameters, with TN denoting the number of true negatives. In our case, however, TN was not accessible, so it was replaced by FDR. Similar to a standard ROC plot, in this representation, tests with better performance are represented by points in the upper left hemisphere of the plot, whereas tests with lower performance yield points in the lower right hemisphere. In Figure 7, we plotted TPR over FDR for various data sets, analyzed with (blue) and without (red) temporal filtering. In general, for all tested parameter settings the temporal filtering method behaves better. The difference becomes substantial in the case of signals of low brightness. The improvement in the modified ROC plot was similar for molecules exhibiting frequent and rare blink events (Appendix A). Moreover, using the temporal filtering method based on Fourier filtering slightly outperformed a median filter (Appendix A). 

The key point of localization microscopy is to achieve a high accuracy in localizing molecules, defined as the distance between the determined position of the detected signal and ground truth. In Figure 8 we compare the positional accuracy with and without temporal filtering, yielding a small reduction of the localization errors upon filtering. 

## 3. Discussion

We presented a new algorithm to pre-process image sequences using Fourier analysis of temporal fluctuations, which helps in identifying blinking single molecule signals over background noise. Upon analysis of the obtained filtered images with standard localization software, we observed improvement in measures such as true positive rates and false discovery rates and essentially unchanged positional accuracy. 

Temporal filtering adds to the spectrum of available filter methods, such as denoising, intensity-thresholding, and constraining the width of the fitted point-spread function, and should be combined with them in practice. It is universally applicable for SMLM analysis independent from the biological or experimental settings: whenever the signal-to-noise ratio is low, temporal filtering will lead to an improvement. As such, it is applicable to any blinking label with low τ_on_/τ_off_ ratio, including besides fluorescent dyes and, for example, gold nanoparticles [24]. The new method is particularly useful for the analysis of rather dim single molecule signals on samples containing high fluorescence background, as it occurs in fungi, but also thick tissue slices or PAINT microscopy. As seen in Figure 7, the performance upon temporal filtering hardly changes over the simulated range of intensities, whereas without filtering there is a clear disadvantage when it comes to dim signal analysis. 

The test has the advantage that it depends only on a single, adjustable parameter, which is the threshold ω_T_. Its choice should account for the blinking rate of the molecules: In principle, “on”-states with a short duration allow for setting a high threshold, which makes them rather easy to identify over background. This particularly relates to scenarios in which molecules show brief “on”-periods separated by long “off”-periods and in which molecules disappear due to photobleaching within the image sequence. In contrast, long-lasting “on” states will be more difficult to discriminate with this approach. Still, the method is surprisingly robust against variation of ω_T_, as long as values above 60% are considered: TPR still increases, and FDR only slightly deteriorates (Figure 6b). 

Experimental settings such as frame rate or illumination time do not directly influence the quality of temporal filtering, but they have an indirect influence by changing the blinking rate, the amount of bleaching, and the total amount of blinks per molecule. Provided that the change of the background is significantly slower than the fluorescent signals, temporal filtering will work with the right choice of ω_T_ as described below. 

Care should be taken, however, in cases were the background strongly fluctuates. While this seems to be exceptional in biological settings, it cannot be fully excluded. We hence recommend obtaining two experimental data sets—a stained and an unstained sample—for comparison. By definition, the unstained sample contains only false positives, whereas the stained sample contains both true and false positives. Beginning with the analysis of the stained sample, one needs to first adjust the parameters for the subsequent single molecule detection algorithms, such as mask size, peak intensity, spatial denoising etc., both for the filtered and the non-filtered scenarios. A good choice for the initial ω_T_ is 80%. Next, using the same parameter settings for the analysis of the unstained sample yields the amount of false positive signals. Ideally, this number should not increase upon temporal filtering. Comparing the signals of stained samples with unstained samples gives an estimation of the true positives. Ideally, upon filtering, the number of true positives increases. Variation of ω_T_ can then be used to optimize the performance. 

Taken together, the new method may help to open up challenging biological systems to super-resolution microscopy. The increased true positive rate may further help to reduce the time required for obtaining an image sequence with the same amount of correct single molecule localizations. As temporal filtering based on Fourier analysis is compatible with most other image processing tools, it represents a versatile new toolkit for single molecule localization microscopy. The improved quality of localization data will ultimately lead to improvements in their subsequent analysis, for example, to achieve best resolution in imaging [6] or to sensitively detect molecular organization in (multi-color) correlation analysis [25,26,27].

## 4. Materials and Methods

*T. atroviride* wild-type (ATCC 74058) and a mutant derived thereof that constitutively expresses an *sfp2-mEGFP* fusion construct [20] were cultivated in sterile glass bottom ibidi 2 well µ-slides (ibidi, Martinsried, Germany). The wells were coated with 1 mL poly-d-lysine (Sigma-Aldrich, Darmstadt, Germany) and incubated at room temperature for 30 min. Afterwards, the solution was removed and the wells were dried in a sterile environment. A plug of seven day-old *T. atroviride* wild-type or *sfp2-mEGFP* mutant culture was applied to the edge of each chamber and 100 µL of potato dextrose broth (PDB) were added. Slides were incubated for five days in complete darkness at 25 °C. Before measurements, samples were fixed with a 4% Paraformaldehyde solution and—in the case of labelled hyphae—stained with an AF647 (Thermo Fisher Scientific, Waltham, MA, USA) conjugated GFP-Trap (Chromotek, Planegg-Martinsried, Germany).

All microscopy experiments were carried out on a custom built super-resolution setup, which was based on an inverted Zeiss Axiovert 200 body equipped with a Zeiss Apochromat 100x/1.45 NA oil-immersion objective (Zeiss, Oberkochen, Germany). Samples were illuminated with a 637 nm Coherent OBIS laser and images were detected on an iXon Ultra 897 EM-CCD camera (Andor, Belfast, UK). Experiments were performed in epi-configuration, using stroboscopic illumination with 1 ms illumination time and 7 ms delay time. For measurements, a dSTORM buffer optimized for AF647 [28] was applied. 

Simulations and data analysis were performed in Python 3 utilizing packages from the SciPy ecosystem, including NumPy, the SciPy library, Matplotlib and pandas. Microscopy images were imported and exported to hard disk with help of the PIMS package.

To generate the ground truth images, we measured five distinct hyphae of unlabeled *T. atroviride* under STORM conditions. For each image sequence, we defined a region of interest (ROI) covering exactly the hyphae regions. We next generated random positions within the ROI at the specified density of molecules. For each simulated single molecule position, we simulated blinking by binary traces representing the “on” and the “off” state of the molecules. Exponentially distributed transition times with mean values of τ_on_ and τ_off_ characterized the duration of the “on” and the “off” state, respectively. To avoid synchronized blinking at the beginning of the sequences, the starting points within the blinking traces were chosen randomly. For each frame, the signals of molecules in the “on”-state were simulated by Gaussians with amplitudes drawn from a lognormal brightness distribution (mean value of B and a $\sigma_{B}=\frac{B}{3}$). We assumed a sigma-width of the simulated Gaussians of 160 nm, reflecting the measured size of single AF647 molecules. Finally, all simulated images were overlaid with the respective image of the recorded background model. 

We used 3D-DAOSTORM [22] and an algorithm published by Gao et al. [23] to analyze the simulated image sequences. Parameter settings were adjusted for each threshold ω_T_ independently. Every detected localization with a counterpart within a radius of 160 nm in ground truth was defined as a true positive, all other detected localizations as false positives. Simulated localizations that were not detected by the algorithm were defined as false negatives. 

For the median filter [19], we used a sliding window with a width of 2000 frames centered at the actual frame, from which we calculated the median per pixel, which was subtracted from the actual image. At the first 1000 and the last 1000 frames, we set the window to the first 2000 or last 2000 frames, respectively. 

## Figures and Tables

**Figure 1 molecules-23-03338-f001:**
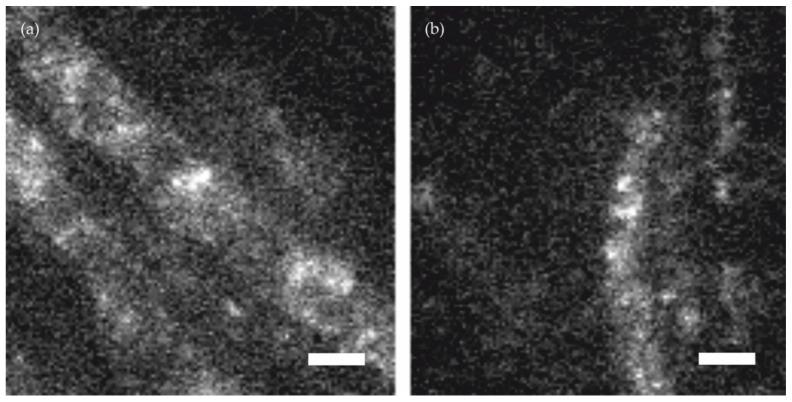
Representative images of *Trichoderma atroviride* hyphae recorded under identical imaging conditions. Panel (**a**) shows unstained wild-type, panel (**b**) an *sfp2-mEGFP* expressing strain labelled with an AF647 conjugated GFP-Trap. Scale bar = 3 µm.

**Figure 2 molecules-23-03338-f002:**
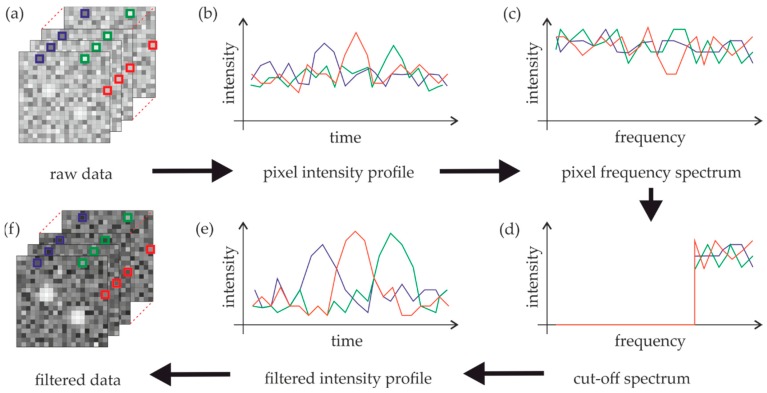
The procedure of temporal filtering. The image sequence containing faint single molecule signals is analyzed on a pixel-by-pixel basis (**a**), yielding the time course of the intensity for each pixel (**b**). We use a fast Fourier transformation (FFT) to obtain the respective frequency spectrum per pixel (**c**). After applying a high-pass filter with a threshold frequency ω_T_ (**d**), an inverse FFT is used to transform the filtered frequency spectra back to time-space (**e**), yielding the final temporal filtered images (**f**).

**Figure 3 molecules-23-03338-f003:**
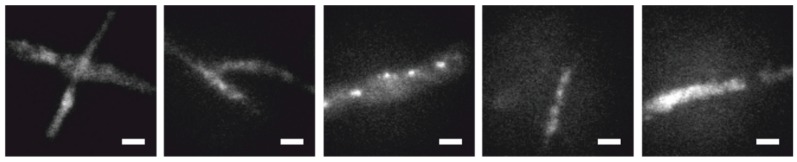
Representative images of *T. atroviride* wild-type hyphae from the five movies used as autofluorescence models. Scale bars = 3 µm.

**Figure 4 molecules-23-03338-f004:**
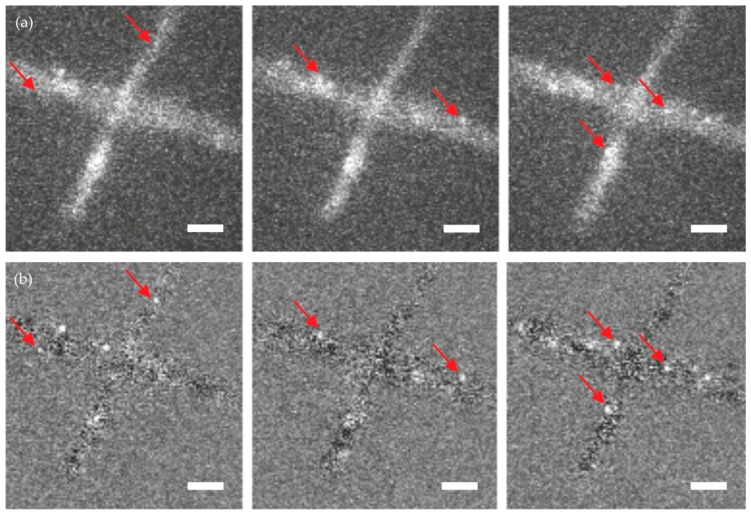
Comparison of *T. atroviride* wild-type images with overlaid simulated single molecule signals. Shown are images without (**a**) and with (**b**) temporal filtering. The red arrows indicate signals that were missed in the initial analysis but were detected after temporal filtering. Scale bar = 3 µm.

**Figure 5 molecules-23-03338-f005:**
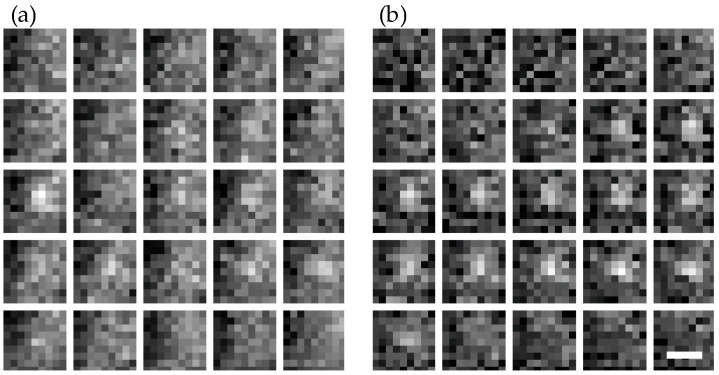
Time trace of a single molecule signal overlaid on a *T. atroviride* wild-type movie. Shown are images without (**a**) and with (**b**) temporal filtering. Scale bar = 0.8 µm.

**Figure 6 molecules-23-03338-f006:**
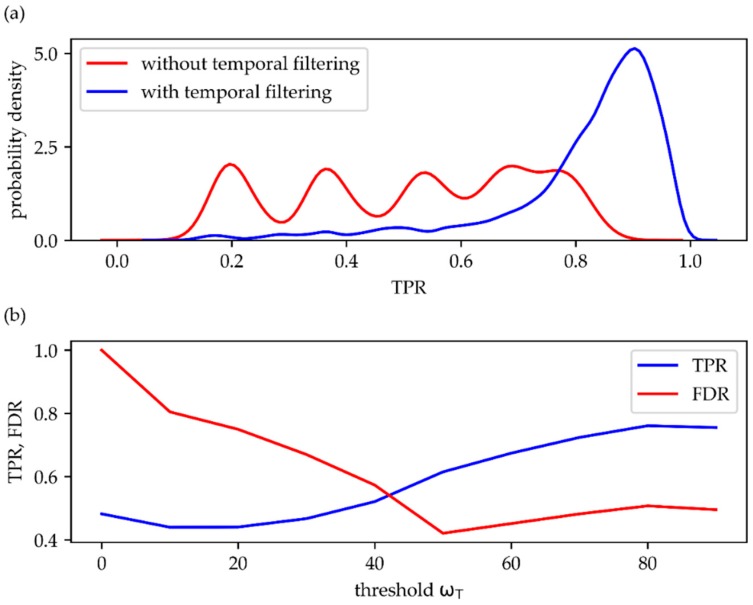
Quantitative comparison of analyzed data without and with temporal filtering. (**a**) The distributions of the obtained TPR without (red) and with (blue) temporal filtering. Data include simulations with varied parameters for B, τ_on_, τ_off_, ω_T_. (b) Plot of the dependence of TPR (blue) and FDR (red) on the threshold ω_T_. Panel (**a**) was analyzed with 3D-DAOSTORM [22], panel (**b**) with the algorithm by Gao et al. [23].

**Figure 7 molecules-23-03338-f007:**
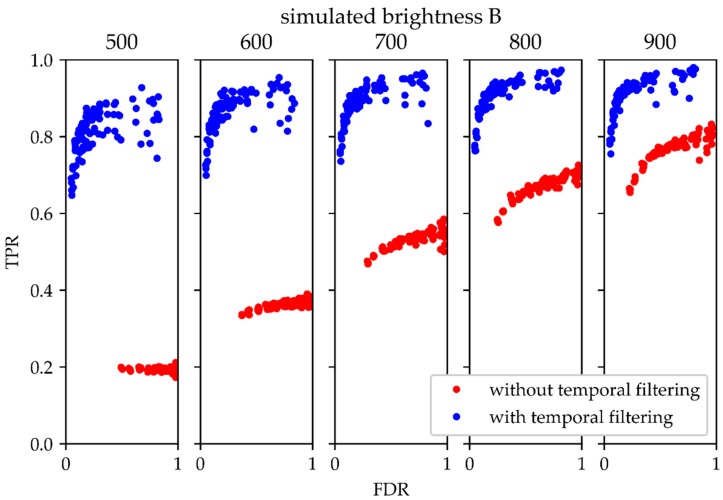
Performance test for the temporal filtering. The true positive rate (TPR) is plotted versus the false discovery rate (FDR) for different simulated single molecule brightness values. The comparison shows data obtained without (red) and with (blue) temporal filtering. In this plot, the top left corner corresponds to the ideal case with no false positive or negative signals, whereas the bottom right corner corresponds to the worst case of only false positive signals. Data include simulations with varied parameters for τ_on_ and τ_off_. ω_T_ was set to 80%, analyzed with the algorithm by Gao et al. [23].

**Figure 8 molecules-23-03338-f008:**
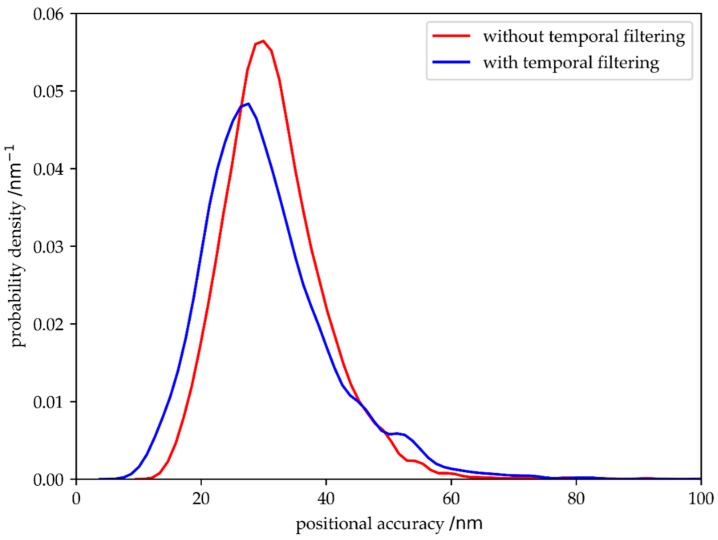
Comparison of the positional accuracy obtained for single molecule signals with (blue) and without (red) temporal filtering. Data include simulations with varied parameters for B, τ_on_, τ_off_ and ω_T_. Data were analyzed with the algorithm by Gao et al. [23].

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
