# Peer review of "Temporal Filtering to Improve Single Molecule Identification in High Background Samples"

_molecules, 2018, doi:10.3390/molecules23123338_

Round 1
Reviewer 1 Report
This work is unfortunately not novel. Temporal filtering based background subtraction in one form or another has been used since the very early days of SMLM (e.g. Baddeley et Al, BiophysJ 2009, & Baddeley et Al, Nano Research 2011 - described in methods section in each case) – although from discussions with colleagues at the time I was under the impression that we were not the only ones to implement it and that it was reasonably common practice. I certainly thought that it was a fairly obvious processing step, but given its omission in many of the more recently published analysis algorithms, I wish I’d given it more prominence. Whilst it could fairly be argued that those original papers did very little to characterize the effect of temporal background subtraction, it has recently been revisited, expanded, and described in much more detail e.g. (Hoogendoom et Al, Sci Rep, 2014 - https://www.ncbi.nlm.nih.gov/pmc/articles/PMC3900998/) and https://www.biorxiv.org/content/early/2018/04/05/29526. There are likely to be other papers which I've missed.
A difference between this work and previous uses of temporal filtering is the decision to use a Fourier domain filter rather than the more common time domain filtering. This is however a minor technical detail which is not well justified. The use of a sharp cutoff in Fourier space can be expected to cause ringing artifacts (due to the fact that it is effectively an IIR filter with a sinc like impulse response), potentially aliasing noise and/or event signals from very distant frames within the sequence into the current frame. In contrast, the established time-domain approaches have a temporally limited impulse response. Fourier domain filtering would also be expected to confer a significantly higher computational cost than time-domain filtering (which, due to the limited duration of blinks, can be performed over a reasonably short time window).
Regardless of the method chosen, temporal filtering alters the noise characteristics of the data, likely invalidating the noise model used for fitting (especially for Poisson based noise models). As a result, preprocessing with a temporal filter and then piping the processed data through an existing algorithm is far from ideal. For this reason, temporal filtering approaches should ideally use a modified fit algorithm which takes the background subtraction process into account. In the interests of space I won’t go into details here, but the authors are welcome to contact me directly at d.baddeley@auckland.ac.nz for information on how we have done this is done in our open-source package. Depending on the application, the penalties for not using a modified fit algorithm can be significant (from personal experience, a suitably modified fit algorithm is critical for high quality 3D localization, but less so for 2D).
There is one minor nugget of novelty in the paper – the characterization of point detection efficiency in terms of a receiver operating curve, both with and without temporal filtering (previous descriptions of temporal filtering have not rigorously quantified the effect). This is an elegant way to assess the effects of different background correction algorithms. That said, detection efficiency is highly dependent on the choice of algorithm, with many possible choices available of which the authors only use DAO-STORM. It is impossible to infer how much of the described effect is specific to use with DAO-STORM.
In addition to the lack of novelty, there are a number of technical deficiencies – e.g. the authors do not describe how negative values arising from the background subtraction process are handled. As far as I can tell, the noise model used in DAO-STORM assumes strictly positive values. If the values are truncated at zero after background subtraction this would have the effect of putting the localizations on an artificial “pedestal” with a value of roughly half the background noise amplitude, which might be expected to bias the fit parameters.
If you wish to salvage something from the paper, I’d recommend doing a thorough comparison of the different methods of temporal background subtraction (time domain vs Fourier domain, linear - i.e. rolling average etc vs non-linear – e.g. median, minimum, etc …) and a number of different localization routines, including ones which take background subtraction into account in their noise models. I would see this as being an entirely new paper, not simply a resubmission.
Author Response
see attached pdf

Reviewer 2 Report
Single Molecule Localization Microscopy (SMLM) has become a powerful
method to study biological systems on the meso- and nano-scale.
Depending on the labelling technique applied and the colour channel detected, intrinsic blinking of of self-fluorescent molecules may lead to an unspecific background that is not easy to discriminate. Therefore the authors have developed a novel filtering approach using the temporal blinking characteristics to separate real signals from background. The method and the tests are well described and the results for the special application are leading to an improvement of a factor 3.5 as compared to the unfiltered data, thereby the localization accuracy is not significantly changing.
The article is well written and I recomment publication after some additional revision.
) The authors very much focus the results on the application of filamentous fungi and typical dyes used there. However, the introduction is in that way misleading that the reader gets the impression that the filtering method is applicable in general. Therefore I would recomment to better express the limits to the special application and discuss other possibilities.
) In SMLM very often blinking dyes only show very few (1-3) blinking events. This should be discussed in more detail.
) In cell biology the self-fluorescence is mostly reduced to the green part of the spectrum. Therefore an alternative procedure may be the application of dyes in the infra-red or red spectrum. This should be discussed.
) On the other hand, labelling by nano-gold particles (see for instance Moser et al. Biophysical Journal, Voll 110, 2016) should be an appropriate labelling scheme for using the presented method, since plasmonic blinking is occuring continuously and the particle size would difine the frequency. This should be discussed or if possible, additional results might be shown with such an approach (only suggested if the laboratory of the authors are able to shortly do the experiments). By the way, using particles as labels has the advantage to control the specificity by electron-microscopy (see for instance Hildenbrand et al. PlosONE, 2018).
) Finally a comparison to other filtering procedures (e.g. intensity thresholding etc.) may be discussed. In addition the application of a-priory knowledge or multi-colour labelling allows to discriminate specific from non-specific signals. This should be discussed with some corresponding citations.
Author Response
see attached pdf

Reviewer 3 Report
The authors present a procedure to partially de-noise SMLM images using high pass filtering of the per-pixel intensity signals in an image sequence. The idea is very interesting, could prove useful in the field. I am happy for this paper to be published. Also, the paper is well written with all parts described and explained clearly. The hope is that more papers nowadays are written in this manner.
I do, however, have some comments and questions, which may require some minor revisions:
1) The authors describe use of simulated signals included on actual auto-fluorescence data. Real signals, however, bleach over time which is a very common artifact. Can the authors add a few sentences to the manuscript describing the possible effects of bleaching to the acquired metrics?
2) The authors should mention come implications of realistic experimental parameters including exposure times, frame rates, number of frames, etc. For example, if the frame rate is 20 frames/ second, how would the overall performance compare with a frame rate of 2 frames/second for data acuiqred over the same time span. How would this impact the choice of the threshold wT , and the resulting performance of the algorithm?
3) In the last sentence of the abstract, the authors mention, “we found an improvement up to a factor 3.5 while …” This sentence should be revised, and it should be made very clear in this sentence what the metric of this improvement is.
Author Response
see attached pdf

Round 2
Reviewer 2 Report
The authors have clearified the different points mentioned in the first review. They expleined in detail what they have done or not and why. I agree with their arguments. The manuscript can be accepted.